# Myocardial Ischemia–Reperfusion Injury: Unraveling Pathophysiology, Clinical Manifestations, and Emerging Prevention Strategies

**DOI:** 10.3390/biomedicines12040802

**Published:** 2024-04-04

**Authors:** Marios Sagris, Anastasios Apostolos, Panagiotis Theofilis, Nikolaos Ktenopoulos, Odysseas Katsaros, Sotirios Tsalamandris, Konstantinos Tsioufis, Konstantinos Toutouzas, Dimitris Tousoulis

**Affiliations:** 1st Cardiology Clinic, ‘Hippokration’ General Hospital, School of Medicine, National and Kapodistrian University of Athens, 11527 Athens, Greece; anastasisapostolos@gmail.com (A.A.); panos.theofilis@hotmail.com (P.T.); nikosktenop@gmail.com (N.K.); odykatsaros@gmail.com (O.K.); stsalamandris@hotmail.com (S.T.); ktsioufis@gmail.com (K.T.); ktoutouz@gmail.com (K.T.); drtousoulis@hotmail.com (D.T.)

**Keywords:** myocardial ischemia–reperfusion injury, myocardial infarction, pathophysiology, endothelium, clinical implications, prevention, treatment

## Abstract

Myocardial ischemia–reperfusion injury (MIRI) remains a challenge in the context of reperfusion procedures for myocardial infarction (MI). While early revascularization stands as the gold standard for mitigating myocardial injury, recent insights have illuminated the paradoxical role of reperfusion, giving rise to the phenomenon known as ischemia–reperfusion injury. This comprehensive review delves into the intricate pathophysiological pathways involved in MIRI, placing a particular focus on the pivotal role of endothelium. Beyond elucidating the molecular intricacies, we explore the diverse clinical manifestations associated with MIRI, underscoring its potential to contribute substantially to the final infarct size, up to 50%. We further navigate through current preventive approaches and highlight promising emerging strategies designed to counteract the devastating effects of the phenomenon. By synthesizing current knowledge and offering a perspective on evolving preventive interventions, this review serves as a valuable resource for clinicians and researchers engaged in the dynamic field of MIRI.

## 1. Introduction

The mismatch between blood flow and tissue demand can result in hypoperfusion and ischemia, leading to irreversible tissue necrosis and organ failure. Tissue hypoperfusion has been observed in several conditions, including acute coronary syndrome (ACS), organ transplantation, and sepsis [1]. For ACS, the ACCF/AHA/SCAI guidelines recommend early revascularization to control myocardial injury and protect the viability of the tissue [2]. Indeed, thrombolytic therapy or primary percutaneous coronary intervention (PCI) is commonly employed and effective in limiting the size of a myocardial infarct with subsequent more favorable clinical results [2]. However, paradoxically, adverse clinical outcomes are sometimes observed after the acute restoration of blood flow, leading to exacerbated tissue damage. Cohort data have indicated that reperfusion has the potential to induce subsequent injury in ischemic tissue, a phenomenon referred to as ischemia–reperfusion injury (IRI). Myocardial ischemia–reperfusion injury (MIRI) is an independent aggravating factor leading to adverse cardiovascular events shortly after myocardial ischemia, cardiac surgery, or circulatory arrest. MIRI is a field of extensive research, given that it can account for up to 50% of the final size of a myocardial infarct in experimental models [3].

The phenomenon is characterized by multi-systemic involvement, and paradoxically exacerbated cell dysfunction and death following reperfusion procedures [4]. This phenomenon affects not only the target organ but also distant tissues, potentially leading to multi-system organ failure [4]. Understanding the mechanics of ischemia–reperfusion has made significant advances using acute myocardial infarction (MI) models. However, the enthusiasm linked to translating these discoveries into patient care is tempered by the debatable outcomes of clinical studies. This review provides a comprehensive overview of recent advancements in MIRI research, summarizing our current understanding of the associated mechanisms and identifying potential targets for therapeutic intervention.

## 2. Myocardial Ischemia–Reperfusion Injury Mechanisms

MIRI was first described during the early 1970s in experimental models of canine hearts. Jennings et al. observed necrotic histological changes during the first 30–60 min after reperfusion with these features normally being met after the first 24 h of a permanent coronary occlusion [5]. Subsequently, numerous studies have endeavored to explore the intrinsic mechanisms and diverse pathological processes involved in MIRI. Anaerobic glycolysis has been identified as the primary source of ATP for a brief period of 15 to 20 s following coronary occlusion [6]. This energy supply persists for approximately 60–90 min, providing basal energy requirement. Depletion of energy resources and the cessation of ATP production led to contracture-rigor in the myocardial muscle in under 5 min [6].

Four types of injury have been described during MIRI: (i) myocardial stunning, (ii) the no-reflow phenomenon, (iii) reperfusion arrhythmia, and (iv) lethal reperfusion injury.

### 2.1. Myocardial Stunning

Myocardial stunning is described as a persistent mechanical heart dysfunction after reperfusion of the occluded coronary artery. This is a fully reversible phenomenon with the recovery of heart muscle lasting for days [7]. Most of the improvement observed within the first week after the MI mainly attributed to the systolic and diastolic function recovery [7,8]. Since myocardial stunning is a time-dependent phenomenon, the extent and duration of blood flow deprivation, myocardial tissue temperature, and left ventricle loading conditions have been closely associated with its severity. From a pathophysiologic aspect, after reperfusion, mitochondrial oxidative phosphorylation rapidly recovers, while more time is required for contractile power to reach pre-ischemic values [9]. The stunned myocardium exhibits a relative excess of oxygen consumption for a given rate of contractile work, leading to decreased mechanical efficiency. This phenomenon can be partially ascribed to the swift restoration of intracellular pH during the reperfusion process [2,10]. Anaerobic glycolysis during the ischemic phase increases intracellular H^+^, which is subsequently transported into the extracellular space in exchange for Na^+^ to normalize pH upon reperfusion. The intracellular Na^+^ activates the sarcolemmal 2 Na^+^/Ca^2+^ exchanger, resulting in exchange of intracellular Na^+^ with extracellular Ca^2+^, leading to calcium overload and cell death [3].

### 2.2. No-Reflow Phenomenon

The no-reflow phenomenon poses a common challenge for physicians and significantly impacts the outcomes of PCI, marked by inadequate microvascular perfusion [11]. This phenomenon occurs in 0.6–3.2% of percutaneous coronary intervention (PCI) cases in the current practice [12]. It is known that the no-reflow phenomenon is linked with delayed functional recovery, left ventricular systolic impairment, heart failure, life-threatening arrhythmias, and death [13]. The main mechanisms contributing to the phenomenon mainly involve endothelial damage, leukocyte plugging, and mechanical compression [14,15]. More particularly, the intracellular changes during MIRI lead to the formation of free radicals or reactive oxygen species (ROS) via the accumulation of H^+^ and Ca^2+^. ROS formations as well as activation of pro-inflammatory pathways seem to be critical contributors to the no-reflow phenomenon in MIRI. Oxygen-derived free radicals, as crucial mediators of MIRI, generate various forms of oxygen species. Reactive oxygen intermediates directly damage cellular DNA, proteins, and lipids while also activating stress response pathways [14,15]. Endothelial damage during MIRI amplifies the infarct size and exacerbates microvascular hypoperfusion, initially localized to the necrotic zone but extending to the subepicardium after prolonged ischemia. Thrombi and leukocyte embolization are additional factors contributing to the no-reflow phenomenon. Finally, myocyte swelling and tissue edema, primarily induced by the release of chemoattractants, result in the accumulation of neutrophils in the infarct zone within the initial 6 to 24 h of myocardial reperfusion, leading to microvascular occlusion through external compression [3,14,15].

### 2.3. Reperfusion Arrhythmia

During the initial 48 h, patients with MI commonly exhibit reperfusion arrhythmias [16]. The most common arrhythmias include an accelerated idioventricular rhythm, atrial fibrillation, sinus bradycardia, non-sustained ventricular tachycardia, and sinus tachycardia [17]. Although the vast majority of them are well tolerated, ventricular tachycardia or ventricular fibrillation are associated with a high risk of mortality. Reperfusion arrhythmias are predominantly observed following a brief ischemic duration, and it is hypothesized that the generation of these arrhythmias is critically influenced by oxygen-derived free radicals [18]. Alterations in intracellular and extracellular levels of calcium, potassium, and sodium during MIRI contribute to the dispersion of cardiomyocyte refractoriness, facilitating the formation of re-entry. Reperfusion arrhythmias are not only driven by the re-entry mechanism but also by triggered activity resulting from early and delayed afterdepolarizations [19,20]. The increased autonomic stimulation of Purkinje fibers near the ischemic region, enhancing automaticity or triggered activity, may result in accelerated idioventricular rhythm, posing a routine challenge for cardiologists. Several factors, including acidosis, α-adrenergic stimulation, and angiotensin II release, heighten the susceptibility of ischemic myocardium to reperfusion arrhythmias [21].

### 2.4. Lethal Reperfusion Injury

Lethal reperfusion injury represents the most severe consequence of MIRI, serving as the primary impediment to myocardial tissue recovery. It is characterized by myocardial damage upon the restoration of coronary blood flow following an ischemic episode, leading to immediate cardiomyocyte death during the early reperfusion phase [22,23]. The fate of the cell is intricately linked to the permeability of the mitochondrial inner membrane during reperfusion: Minimal permeability allows cell recovery, moderate permeability leads to programmed cell death, and severe permeability results in cell necrosis due to inadequate energy production [24]. Not one but multiple programs of cell death, including apoptosis, necrosis, necroptosis, and autophagy-associated cell death, are activated during ischemia and reperfusion [25]. Necrosis is marked by cellular and organelle swelling, followed by the rupture of surface membranes and the release of intracellular contents. Importantly, necrotic cells act as robust stimulants of the immune system, frequently triggering extensive inflammatory cell infiltration and cytokine production in the affected region. This cascade may exacerbate the severity of MIRI [26,27]. Necroptosis, a type of cell death that shares features with both necrosis and apoptosis, is recognized as a regulated process. Lysosomal degradation and increased expression of autophagy-related genes characterize autophagy, which is considered an adaptive response to sublethal stress. Despite the involvement of various cell death mechanisms in lethal reperfusion injury, necrosis remains the predominant mediator of cardiomyocyte death during the MIRI process [26,27].

Metabolic studies also highlight the critical role of the mitochondrial permeability transition pore (PTP), a non-selective channel of the inner mitochondrial membrane in the occurrence of lethal reperfusion injury [28]. Notably, the PTP is observed to remain closed during ischemia, only opening within the initial minutes after reperfusion, induced by factors such as mitochondrial Ca^2+^ overload, oxidative stress, restoration of a physiological pH, and ATP depletion. Consequently, modulation of the PTP presents a prospective and innovative therapeutic target for preventing reperfusion injury in the future [8,28].

## 3. The Role of Endothelium in MIRI

Ischemia–reperfusion induces vascular endothelial dysfunction through various mechanisms, encompassing cytotoxicity induced by alterations in pH, oxidative stress resulting from the excessive generation of ROS, and inhibition of endothelial nitric oxide synthase (eNOS) and nitric oxide (NO) [29,30]. Recent studies have provided additional insights into the molecular mechanisms of endothelial I/R injury, involving the modulation of ion channels and gap junction proteins. Kumar et al. have shown that myocardial ischemia is associated with acidosis, and not anoxia, which provokes a Ca^2+^ leak from the endoplasmic reticulum and the activation of caspase-12 and caspase-3 [31]. Acidic preconditioning, characterized by the upregulation of the antiapoptotic protein Bcl-xL, serves as a protective mechanism against ischemic apoptosis in coronary endothelial cells [32]. Furthermore, extracellular acidosis exerts a pronounced inhibitory effect on Ca^2+^ entry into endothelial cells, consequently suppressing the production of vasoactive substances, thereby potentially influencing I-R-induced endothelial dysfunction.

ROS, generated abundantly by cardiomyocytes, coronary vascular endothelium, and inflammatory cells during MIRI, contribute to intravascular microthrombosis, reduced blood flow, and activation of inflammatory cells. Oxidative stress-induced activation of endothelial cells promotes the expression of adhesion molecules, such as E-selectin, P-selectin, and intercellular adhesion molecules (ICAMs), facilitating the recruitment of neutrophils [33,34]. The pivotal role of nuclear factor kappa-B (NF-κB) in I-R-induced endothelial cell activation involves tyrosine phosphorylation of IkBa induced by oxidative stress, leading to NF-κB nuclear translocation and subsequent transcriptional activation of proinflammatory, procoagulant, and vasoactive genes [35]. Additionally, oxidative stress activates mitogen-activated protein kinases (MAPKs), influencing NF-κB transactivational activity [34,36]. Endothelial cells not only serve as targets but also as sources of ROS, contributing significantly to vascular dysfunction after I-R [37]. MIRI leads to an increase in endothelial permeability, attributed to ROS released from activated leukocytes, causing alterations in endothelial cytoskeletal structures and promoting intercellular gap formation [38]. Simultaneously, re-energization-induced activation of endothelial contractile machinery contributes to endothelial barrier failure, facilitating the migration of neutrophils and other inflammatory cells into the injured myocardial tissue, thereby exacerbating MIRI [39]. Furthermore, IRI disrupts the balance between endothelium-derived constricting and relaxing factors, leading to interrupted blood flow and organ perfusion. Increased production of vasoconstrictors, such as endothelin-1, and reduced availability of endothelium-derived relaxing factors, particularly NO and endothelium-derived hyperpolarizing factor (EDHF), contribute to disturbances in blood flow during myocardial ischemia [34,40,41,42].

The decrease in NO bioavailability, a well-known consequence of MIRI, involves multiple mechanisms, including eNOS inhibition, arginase activation, and increased ROS production. Uncoupling of eNOS during MIRI shifts its function from NO production to the generation of ROS [43,44]. Reduction in NO and increased ROS production worsen endothelial IRI, impacting cardiac output by reducing the contractility of actin–myosin fibers within myocytes [45]. The role of EDHF in vasodilation becomes increasingly significant as vessel diameter decreases [46,47]. The conventional EDHF pathway includes the opening of intermediate and small conductance Ca^2+^-activated K^+^ channels (IKCa and SKCa, respectively) on the plasma membrane of endothelial cells [48]. This phenomenon leads to conductible hyperpolarization via myoendothelial gap junctions and K^+^ efflux-mediated hyperpolarization through activation of inwardly rectifying K^+^ (Kir) channels and Na^+^-K^+^-ATPase on adjacent smooth muscle cells [49]. In some types of vessels, such as coronary arteries, an atypical EDHF response mediated by epoxyeicosatrienoic acids (EETs) has been proposed [50].

While animal models suggest potentiation of the EDHF-type response in MIRI, contradictory evidence indicates compromised EDHF function under IRI conditions. For instance, exposure to hypoxia/reoxygenation (H/R) significantly attenuates EDHF-mediated relaxation in porcine coronary arteries and blunts EDHF response in coronary microveins. This impairment of EDHF responses is associated with the inhibition of IKCa and SKCa currents in coronary endothelial cells during H/R exposure [51,52,53] (Figure 1).

## 4. Preventive and Therapeutic Approaches

In recent decades, MIRI has become a subject of extensive research, focusing not only on understanding the involved pathophysiological pathways but also on developing strategies for its prevention. Numerous suggested pharmacological and non-pharmacological modalities have been proposed, and ongoing trials are diligently assessing their efficacy and safety (Table 1).

### 4.1. Ischemic Preconditioning

The phenomenon known as “the cardiac warm-up phenomenon” has been observed in patients with MI who experienced at least one episode of prodromal angina, presenting less severe ischemic damage and a reduction in infarct size [54]. This phenomenon describes how prior brief intermittent periods of myocardial ischemia, such as unstable angina, significantly diminish the size of the infarct in subsequent total occlusion events [55]. The cardioprotective effects of preconditioning is observed only if the reperfusion strategy is applied within the first 3 h. When the sustained coronary occlusion is not followed by reperfusion within that time frame, the protection of conditioning is lost [55]. Notably, when the time interval between the brief protection-inducing coronary occlusion and the infarct-inducing sustained coronary occlusion was extended from 5 min to 2 h, the protection was largely attenuated [56]. Conversely, few studies have reported long-lasting cardioprotective effects of preconditioning when initiated 24 h after application, lasting for about 1–3 days [57]. Despite reducing reperfusion injury and its systemic consequences, direct ischemic preconditioning has limitations due to direct stress on the target organ and mechanical trauma to major vascular structures, restricting its clinical application. Additionally, preconditioning cannot be clinically applied in acute MI cases since the coronary artery is already occluded at the time of hospital admission, and the consequences of ischemic events are often unpredictable. Preconditioning can be used in patients with pre-infarction angina during elective PCI or those undergoing coronary artery bypass grafting [58,59].

### 4.2. Ischemic Postconditioning

Due to the limitations associated with the use of preconditioning, particularly its restricted application in acute MI cases, researchers have explored the postconditioning method. In contrast to preconditioning, the experimental design of postconditioning theoretically enables direct application in clinical settings, especially during PCI. Postconditioning was first introduced in canine hearts by Zhao et al. and is defined as brief coronary reocclusion–reperfusion applied early during myocardial reperfusion following sustained coronary occlusion [60]. This can be achieved by inflating and deflating an angioplasty balloon after reopening the occluded artery. Several studies have demonstrated remarkable results, with a reduction in infarct size of approximately 35%, comparable to the preconditioning method [61]. Significant reductions in edema, and coronary microvascular obstruction, along with satisfactory improvement in left ventricular contractile function, have been observed in patients presenting with acute MI [62,63].

### 4.3. Remote Ischemic Conditioning

Remote ischemic conditioning is a systemic phenomenon where ischemia followed by reperfusion of one organ is believed to protect remote organs [64]. This protection may occur through the release of biochemical messengers into the circulation or activation of nerve pathways, leading to the release of protective messengers. Remote ischaemic conditioning can be induced before coronary occlusion (preconditioning), during coronary occlusion (perconditioning) and after coronary occlusion (postconditioning). It seems the most attractive cardioprotective modality and is currently elaborated in patients with acute MI due to the non-invasive nature of the technique [65].

The pathophysiological pathways involved in remote ischemic conditioning are suggested to be similar to those in preconditioning and postconditioning, with one significant difference. Remote ischaemic conditioning involves the additional signal transfer from the remote tissue, where the protection is initiated, to the heart, where the protection is executed [66,67]. More particularly, the neuronal pathway (peripheral sensory nerves, the spinal cord, the brainstem, and efferent vagal nerves) interacts with humoral factors during remote ischemic conditioning [68]. The spleen is an important relay organ secreting a humoral cardioprotective factor in response to vagal nerve activation. Researchers have identified stimuli in peripheral arteries that activate both neuronal and humoral pathways. Vagotomy, splenectomy, and splenic denervation abrogate the cardioprotective effects of the technique. The use of remote ischemic conditioning has demonstrated a reduction in myocardial infarct size and as indicated by secondary retrospective analysis, improved outcomes in patients with acute MI [66,67,68].

### 4.4. Hypothermia and Vagal Stimulation

Small changes in the range of normothermia have been evaluated as beneficial, reducing the infarct size and coronary microvascular injury in experimental models [69]. Hypothermia has to be achieved during the ischemia phase to provide its cardioprotective effect triggering survival pathways. Unfortunately, a significant reduction in infarct size has not been demonstrated in clinical trials on therapeutic hypothermia in small cohorts of patients with STEMI, potentially due to challenges in achieving rapid and sufficiently intense cooling [69,70,71].

Electrical stimulation of efferent vagal nerves performed during ischaemia or just before reperfusion seemed to reduce the infarct size in rats [72]. Initiated upon hospital admission, low-level electrical transcutaneous stimulation at the right auricular tragus in patients with acute MI led to a reduction in infarct size, minimized arrhythmias, and enhanced ventricular function compared to a sham procedure [72].

### 4.5. Pharmaceutical Treatment

IRI is frequently met after myocardial infarction and its severity is associated with the microvascular obstruction and size of an infarct. Thus, the prevention of IRI may be beneficial for such patients providing a real clinical impact for the management of patients with coronary artery disease. Against this background, several basic and clinical studies have been conducted, aiming to limit this phenomenon, targeting the pathophysiological mechanisms, which were previously described. 

First of all, the role of calcium has been acknowledged in the IRI and has been targeted; however, only preclinical studies have shown the benefits. In experimental studies, the systematic administration of an antagonist of the sarcolemmal Ca^2+^ ion channel, the mitochondrial Ca^2+^ uniporter, or the sodium–hydrogen exchanger showed promising results. Herzog et al. have shown that diltiazem administration prior to reperfusion in Yorkshire swine significantly reduced infarct size (0.13 ± 0.06 g/kg versus 0.42 ± 0.04 g/kg, *p* = 0.01) [73]. The previous results were confirmed by another study evaluating the role of labedipinedilol-A, a dihydropyridine-type calcium channel blocker, in rats [74]. Similar findings were observed with the treatment with other calcium channel blockers, such EGTA (ethylene glycol bis (2-aminoethyl ether)-N, N, N′, N′-tetraacetic acid), nifedipine, and verapamil, in porcine models [75]. Except for calcium channel blockers, methadone or morphine administration has also reduced infarct size in a rat model [76]. Other agents targeting calcium, like magnesium therapy, oxygenation, metabolic modulation with glucose, insulin, and potassium as well as hypothermia, have been explored, showing conflicting results [77,78].

Levosimendan, an inotropic agent, acts by sensitizing the myocardium to calcium effects without increasing the levels of calcium in the myocardial cells. Secondarily, levosimendan opens ATP-sensitive potassium channels located on the plasma membrane and enables myocardial mitochondrial ATP-sensitive potassium channels. Both mechanisms fasten recovery from MIRI [79,80]. Ali Kiraz et al. have shown the benefit of levosimendan administration in limiting myocardial necrosis, inflammation, and edema following edema [81]. Recently, the coadministration of exenatide with levosimendan provided additional benefits in cardioprotection against MIRI in experimental models [82]. Despite the established role of levosimendan in acute heart failure management, it is poorly studied in human subjects regarding MIRI prevention.

IRI has been associated with the activation of inflammatory pathways; thus, several anti-inflammatory agents have been administered for IRI incidence reduction. Interventions aimed at neutrophils during reperfusion, including leukocyte-depleted blood, antibodies against cell-adhesion molecules (P-selectin, IL-6, CD11/CD18, and intercellular adhesion molecule 1), and pharmacologic inhibitors of complement activation, have effectively decreased infarct size in certain experimental studies [78].

During IRI, elevated P-selectin expression from platelets is observed. It has been shown that either transfusing P-selectin knockout mice platelets or administrating a P-selectin inhibitor, such as fucoidan, or blockading the connection between the P-selectin of platelets with the P-selectin glycoprotein ligand-1 (PSGL-1) of leucocytes, a reduced IRI and infarct size are reported in experimental models [83,84,85]. Two trials of phase II evaluated the role of inhibition of P-selectin so as to improve the outcomes of patients presenting with myocardial infarction. The administration of a P-selectin antagonist (soluble form of PSGL-1 fused to the Fc portion of human IgG1) intravenously standard additional thrombolytic therapy failed to show any benefit in patients presenting with STEMI suffering from chest pain for less than six hours; possibly, this trial was discontinued due to the lack of benefit from the treatment [86]. Current data from the investigation of inclacumab, a humanized anti P-selectin-1 antibody, showed that inclacumab administration, compared to placebo significantly reduced myocardial damage, as it is estimated from a significant decrease in troponin I, creatinine, creatine kinase–myocardial band, and peak troponin I after angioplasty [87,88]. However, a phase III clinical trial about the cardioprotective role of inclacumab was never conducted, and inclacumab was not used in clinical practice [89].

The role of interleukin subtypes, like IL-1 and IL-6, has been identified in plaque deterioration and myocardial remodeling after acute myocardial infarction [90]. The levels of interleukins in the blood may play a prognostic role; for example, high levels of soluble IL-6 receptors during the acute phase of STEMI were associated with an increased incidence of MACE [89]. Thus, interleukin subtypes may act as possible targets for pharmacotherapy of IRI prevention. Tocilizumab is an IL-6 receptor antagonist, which is studied extensively in IRI prevention. After a phase II trial showing that tocilizumab decreases the level of CRP and troponin T levels, ASSAIL-MI sought to assess the effect of the tocilizumab on myocardial salvage in acute STEMI by randomizing 199 patients to tocilizumab or placebo and evaluating the myocardial salvage index as measured by magnetic resonance imaging after 3 to 7 days [91,92]. Indeed, tocilizumab increased myocardial salvage in patients with acute STEMI, creating new prospects for the management of ACS and IRI [92]. Moreover, IL-1 has also been recognized as a pharmaceutical target for multiple diseases, including ACS. Anakinra, an IL-1 receptor antagonist, has significantly reduced hsCRP in patients with NSTE-ACS and STEMI in two randomized studies compared to placebo [93,94].

Colchicine is another anti-inflammatory drug used for several diseases; nevertheless, its cardioprotective role has been recently recognized. Its properties are attributed to its effect on macrophages, neutrophils, and endothelial cells; colchicine inhibits the production of matrix metalloproteinase, tumor necrosis factor-α, and other proteolytic enzymes and promotes the production of anti-inflammatory mediators like IL-10 and TGF-β. In patients presenting with STEMI, colchicine significantly reduces creatine kinase-MB and infarct size assessed by CMR [95]. Additionally, COLCOT trial investigators assessed low-dose colchicine versus placebo in patients with previous STEMI or NSTEMI. During an approximately two-year follow-up, patients treated with colchicine suffered from significantly less MACE compared to the placebo arm [96].

Moreover, antiplatelet agents, which are a first line treatment for ACS and CCS, might play a significant role in IRI prevention. The beneficial role of antiplatelet therapy in CAD is well-established in CAD, as it inhibits ADP-P2Y12 receptors and prevents thrombus formation [97]. Several studies have shown that antiplatelet agents may have an impact on IRI reduction. Roubille et al. showed that patients with STEMI had a reduced infarct size, probably as a result of clopidogrel administration [98]. Initially, it was considered that their protective role was associated exclusively with antiplatelets’ antiaggregatory properties. However, newer studies have shown that is probably independent; Yang et al. [99] showed that both clopidogrel and cangrelor have a protective role in rabbits’ hearts against infarction, without mediating aggregatory properties. The authors hypothesize that these results are probably attributed to signal transduction during reperfusion, and administration of these two P2Y12 inhibitors activates the ischemic postconditioning protective signaling mechanism, preventing IRI [99]. Compared to clopidogrel, cangrelor administration in STEMI patients undergoing angioplasty has been more beneficial in both hard clinical endpoints and MVO prevention, which is probably explained by cangrelor’s beneficial impact on cardiomyocytes [100]. For cangrelor, the presence of platelets seems mandatory for preventing IRI; in thrombocytopenic animal models, cangrelor administration failed to reduce infarct size [101]. Similar results were observed in another experimental model using ticagrelor instead of cangrelor, emphasizing the role of platelets [102]. In an experimental model, Vilahur et al. showed that ticagrelor, besides its antiplatelet activity, promotes cardioprotective effects by limiting necrotic injury and edema formation via adenosine-dependent mechanisms [103].

Such findings were confirmed in human subjects by PLATO and HEALING-AMI trials. The PLATO trial compared ticagrelor versus clopidogrel in ACS patients. At one year, the prespecified primary composite endpoint, which included cardiovascular mortality, MI, or stroke, occurred in significantly less patients treated with ticagrelor than clopidogrel (HR: 0.84, 95% CI, 0.77–0.92; *p* < 0.001) [104]. HEALING-AMI included patients with primary PCI-treated STEMI and were randomized to receive ticagrelor-versus clopidogrel-based DAPT and showed that ticagrelor was superior to clopidogrel with regard to left ventricle remodeling, as it was evaluated via a relative change of LVEDV volume measured on 3D echocardiogram and levels of N-terminal pro-B-type natriuretic peptide level [105]. Additionally, ticagrelor prevented endothelial cell apoptosis caused by hypoxia stress; this could be attributed to its effect on the adenosine signaling pathway [106].

Tirofiban, an intravenous GPIIb/IIIa inhibitor, might have a role in IRI prevention. Experimental studies have shown that its administration reduces infarct size, improving microvascular flow after provoked IRI [107]. This is probably associated with the decreased number of platelets and leucocytes infiltrating the infarcted myocardium [108].

Additionally, there are two more antiplatelet agents that have shown promising results in IRI prevention in experimental models: tandem single-chain antibody (Tand-scFvSca-1 + GPIIb/IIIa) and revacept. The first one is a bispecific antibody against both the active substance of GPIIb/IIIa and stem cell antigen-1 receptor. Its administration in mice models sustaining an MI has been associated with the successful delivery of eripheral blood mononuclear cells and platelet-targeted induced vascular progenitor cells to infarcted myocardium, resulting in MI size reduction, increased revascularization, less fibrosis, and improved cardiac viability [109,110].

After the promising results of revacept, a soluble GPVIb-Fc, in experimental models, where its administration showed retained cardiac function and reduced infarct size, ISAR-PLASTER Phase 2 evaluated its safety and efficacy in human subjects [111,112]. Compared to placebo, revacept did not reduce myocardial injury in patients suffering from CCS undergoing elective PCI [113].

Moreover, there are a number of agents focusing on other targets of IRI that are currently under pre-clinical investigation. First of all, VX-765, an inhibitor of caspase 1, reduced infarct size in rat hearts without having an additional beneficial action together with ischemic preconditioning [114]. When VX-765 is administered with ticagrelor or cangrelor in animal models, it provides a further, sustained reduction of MI size; thus, it is an additional promising therapy for patients suffering from MI alongside P2Y12 receptor antagonists [115,116].

Additionally, a number of agents targeting mitochondrial permeability transition pores have been developed and investigated in experimental and human studies. Several drugs, such as cyclosporine, sanglifehrin, delcasertib and elamipretide, targeting mitochondrial permeability transition pores have shown impressive results in experimental studies; nevertheless, their utilization ended in clinical trials, as they failed to show any benefit in multiple studies [117,118,119].

Moreover, angiopoietin-like 4 (ANGPTL4), which inhibits the activation of lipoprotein lipase, an enzyme with well-known anti-triglyceride action, has been studied as another possible pharmaceutical target. Indeed, Dewey et al. have shown that monoclonal antibodies inhibit ANGPLT4 in mice and monkeys and reduce triglyceride serum levels [120]. Although no clear association between ANGPLT4 inhibition and IRI prevention has been found, it is believed that it may play a crucial role in IRI management, and further studies will shed light on this agent [121,122] (Figure 2).

## 5. Conclusions and Future Directions

At present, numerous studies are attempting to alleviate the phenomenon, primarily conducted at rodent levels, with only a limited number extending to non-human mammalian models. The primary constraints associated with drug utilization in MIRI pertain to issues such as the drug’s biodistribution, unfavorable biotransformation, short half-life, and insufficient tissue target specificity [123,124]. However, the rapid advancements in drug delivery systems, particularly nanocarrier systems, hold promise for directly delivering therapeutics to the affected sites in the injured myocardium, at the right time and dose. However, the safety and biocompatibility of emerging nanoparticles need further evaluation and optimization. It is crucial to address issues related to nanoparticle distribution, including size, morphology, and surface modification, with a focus on potential drug accumulation in the liver and kidneys [125]. Cell therapy and stem cell transplantation constitute another revolutionary option in the field [126]. Despite their well-established safety profile, the effectiveness of these approaches, even with direct intracardiac injection, remains under investigation. The challenge arises from the observation that a substantial portion of transplanted stem cells is lost shortly after injection, failing to effectively enhance neovascularization and suppress apoptosis and fibrosis as initially anticipated [127]. Potential solutions to these challenges are foreseen in the near future through the development of innovative drug delivery systems that leverage the intrinsic properties of vectors, such as chemoattractant inflammatory cells, other cell membranes, and ligands [128,129]. Hence, the future prospects for developing drug delivery systems targeting MIRI mitigation appear promising, with a crucial focus on achieving low toxicity and high specificity.

MIRI continues to pose a challenge in medical practice, particularly for interventional cardiologists. Numerous pathophysiological mechanisms, predominantly emphasizing the crucial role of the endothelium, have been suggested. Despite extensive research and numerous experimental and clinical studies on MIRI prevention, no specific agent has found routine clinical application. Additional studies exploring agents that target the underlying pathophysiological mechanisms at both basic and clinical levels are needed to identify optimal treatments for this complex phenomenon.

## Figures and Tables

**Figure 1 biomedicines-12-00802-f001:**
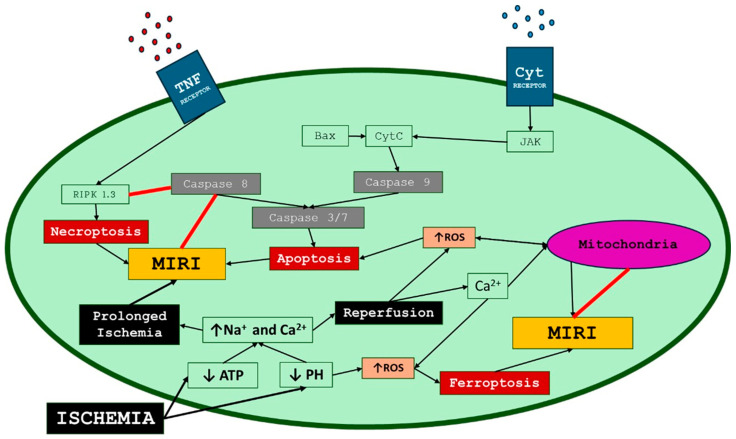
Molecular mechanisms contributing to MIRI. ROS, reactive oxygen species; MIRI, myocardial ischemia/reperfusion injury; CytC, cytochrome c; TNF, tumor necrosis factor; RIPK, receptor-interacting protein kinase; JAK, Janus kinase.

**Figure 2 biomedicines-12-00802-f002:**
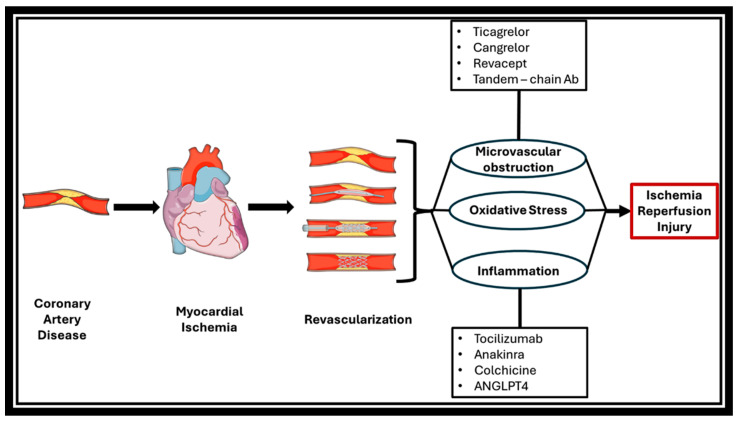
Overview of the pathophysiology behind myocardial ischemia–reperfusion injury (MIRI) and potential pharmaceutical therapeutic targets.

**Table 1 biomedicines-12-00802-t001:** Interventions under investigation according to the procedure phase.

Before Coronary Occlusion	During Coronary Reperfusion	After Coronary Reperfusion
Mechanical Preconditioning	Mechanical Remote Conditioning	Mechanical Postconditioning
Mechanical Remote Conditioning	Hypothermia	Mechanical Remote Conditioning
Infusion of Nitrates	Vagal Stimulation	Regulation of Circadian Rhythm
TLR4 inhibitors	Infusion of TLR4 inhibitors	Erythropoietin
Infusion of Adenosine	Infusion of Adenosine	Epoetin Alpha
Etomoxir	Glucose–Insulin–Potassium (GIK)	
Trimetazidine	Acetaminophen	
Ranolazine		
Dichloroacetate (DCA)		
Levosimendan		

## Data Availability

Upon request to the authors M.S., A.A.

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
