# Peer review of "Myocardial Ischemia–Reperfusion Injury: Unraveling Pathophysiology, Clinical Manifestations, and Emerging Prevention Strategies"

_biomedicines, 2024, doi:10.3390/biomedicines12040802_

Round 1

Reviewer 1 Report

Comments and Suggestions for Authors

The authors put together a comprehensive overview of recent advancements in myocardial ischemia reperfusion injury research, summarizing current understanding of the associated mechanisms and identifying potential targets for therapeutic intervention.

In general, the authors have provided a concise but detailed review of this topic for use of clinicians or researchers in this field.

Therefore, methodology and reliability of data and results is not relevant here. My main criticism has to do, basically, with grammar and style. Once these are addressed appropriately the paper is worth publishing. Following I provide an assortment of these issues and suggest modifications for consideration of authors.  

Comments

Line 31: “…guidelines recommend early revascularization to control myocardial injury and pertain the viability of the tissue.[2] In my opinion, the word “pertain” is misused. Do the authors mean to say “conserve” or “protect”?

Line 56: “Only 15 to 20 seconds after coronary occlusion, anaerobic glycolysis stand as the only significant source of new high-energy phosphate, ATP.[6] This energy supply persists for approximately 60-90 minutes, fulfilling only the fundamental energy requirements…..” A complicated sentence, perhaps “providing basal energy requirement”?

Line 68: “The most improvement of systolic and diastolic function has been observed within the first week after the MI.” Should be “Most of the improvement is observed within…”

Line 91: “More particularly, the intracellular changes during MIRI lead to the formation of free radical or reactive oxygen species (ROS) via the accumulation of H+ and Ca=2. Should be “radicals” and “Ca+2

Line 105: “Patients with MI are slightly common to present at least a reperfusion arrhythmia 105 during the first 48 hours.” Probably “Patients with MI commonly present with…”

Line 219: “The cardioprotective effects of pre-conditioning exert only if reperfusion strategy is applied within the first 3 hours.” “are observed” or “are realized…” instead of “exert”.

Line 231: “Preconditioning can be considered for use in patients with pre-infarction angina during elective PCI or those undergoing coronary artery bypass grafting. “…can be considered for use…”, very awkward. Why not “can be used...”.

Line 277: “Electrical stimulation of efferent vagal nerves, performing during ischaemia or just before reperfusion seem to reduce infarct size in rats.” Should be “performed”.

Line 318: “intravenously additionally standard thrombolytic therapy failed to show any benefit in patients presenting”. “…standard additional thrombolytic therapy failed to…”.

Line 320: “thus, this trial discontinued due to the…..”. Possibly, ”…this trial was discontinued…"

Line 354: “First of all, it is well established the beneficial role of antiplatelet therapy in CAD, as it inhibits ADP-…..”. Better “…the beneficial role of antiplatelet therapy in CAD is well established…

Line 362: “The authors supported that these results are probably attributed to signal transduction during reperfusion and administration of these two P2Y12 inhibitors activate ischemic postconditioning protective signaling mechanism,”. “assume” or “hypothesize that these results are attributed to….”

Line 380: “…versus clopidogrel-based DAPT and showed that ticagrelor was superior to clopidogrel for left ventricle remodeling, as it was evaluated via a relative change of LVEDV volume measured on 3D echocardiogram and levels of N-terminal pro-B-type natriuretic peptide level. Not “for left ventricular remodeling..,”, perhaps “with regard to” or “”as its effect on left ventricular remodeling..” are better.

Line 388: “with the decreased number of platelets and leucocytes infiltrating in infarcted myocardium. Should be “with the decreased number of platelets and leucocytes infiltrating the infarcted myocardium.”

Line 390: “Additionally, there are two more antiplatelet agents having shown promising results” Should be “there are two more antiplatelet agents having showing promising results”.

Line 393: “The first one has a bispecific antibody against both the active….”. Should be “…The first one is a bispecific antibody against…..”

Line 393: “Its administration in mice models suffering from MI has been associated with successful delivery of eripheral blood mononuclear cells and platelet-targeted induced vascular progenitor cells”. “…Its administration in mice models sustaining a MI has been…” probably sounds better.

Line 407: “….size; thus, it is a promising, additional therapy for patients suffering with MI alongside with P2Y12 receptor antagonists.” Probably “thus, it is an additional promising therapy….”

Comments on the Quality of English Language

The paper needs extensive editing and grammar correction. I tried to point some or cite many of these issues.

Author Response

Dear Reviewer,

We are grateful for the feedback, which was very helpful in order to improve the quality of our manuscript. We have tried to address each comment sufficiently in a point-by-point response below:

  • My main criticism has to do, basically, with grammar and style. Once these are addressed appropriately the paper is worth publishing.

Authors’ Response: We apologize for any confusion and, in response, thoroughly reviewed the manuscript for grammar issues. We have addressed and corrected any grammar inconsistencies based on your suggestions.

Reviewer 2 Report

Comments and Suggestions for Authors

I read with interest the paper "Myocardial Ischemia-Reperfusion Injury: Unraveling Pathophysiology, Clinical Manifestations, and Emerging Prevention Strategies". The aim of the paper is appropriate for the SI however some edits are required 

1) Please add more tables and figures to explain some molecular mechanisms of the ischemia-reperfusion injury phenomenon

2) Add more details on the role of some drugs (particularly levosimendan) in the prevention of ischemia-reperfusion injury 

3) Add a section with the research gaps and ongoing trials on the ischemia-reperfusion injury phenomenon

Comments on the Quality of English Language

Minor editing of English language required

Round 2

Reviewer 2 Report

Comments and Suggestions for Authors

The authors adressed my comments. No other  comments.